# A 28 Day Clinical Assessment of a Lactic Acid-containing Antimicrobial Intimate Gel Wash Formulation on Skin Tolerance and Impact on the Vulvar Microbiome

**DOI:** 10.3390/antibiotics9020055

**Published:** 2020-02-01

**Authors:** Elizabeth Bruning, Ying Chen, Karen A. McCue, Joseph R. Rubino, Jeremy E. Wilkinson, Alan D. G. Brown

**Affiliations:** 1Reckitt Benckiser LLC, Montvale, NJ 07054, USA; ilzebruning@msn.com (E.B.); kamccue83@gmail.com (K.A.M.); 2RTL Genomics, Research and Testing Laboratory, Lubbock, TX 79424, USA; jeremy.wilkinson@okstate.edu; 3Alba Science Ltd., Edinburgh EH1 3RH, UK

**Keywords:** feminine hygiene, feminine gel wash, lactic acid, vulvar skin pH, vulvar microbiome, skin microflora, vulvovaginal environment, bacterial 16S rRNA gene, fungal ITS

## Abstract

While intimate feminine hygiene products are widely used as part of daily cleansing routines, little is known about how these products impact the vulvovaginal area and its microbiome stability. This 4 week clinical study assessed tolerance of a novel gel wash containing lactic acid (pH 4.2) for external daily use when used on the external genital area and its effects on skin moisturization, vulvar skin pH, and the vulvar microbiome. After a 7 day pre-study conditioning period, 36 healthy females in three balanced age groups (18–29, 30–44, and 45–55 years) used the gel wash to cleanse their external genital area (mons pubis and vulva) and entire body at least once per day for 28 days. Skin tolerance of the gel wash was assessed by the gynecologist. Effects of the gel wash on vulvar skin microbiota were studied by performing bacterial 16S rRNA and fungal internal transcribed spacer (ITS) microbial richness and diversity analysis. Based on gynecologic assessment after 28 days of use, the gel wash showed acceptable tolerance, with no signs of increased dryness, redness, edema, itching, stinging, or burning. Use of the gel wash was associated with a significant increase in both short-term (single application) and longer-term (daily use for 28 days) skin moisturization. There was no significant change in vulvar skin pH over time with daily product use, and the gel wash did not significantly affect the natural vulvar microbiome species richness or diversity for bacteria or fungi. Results showed that this gel wash is a mild, moisturizing cleanser that maintains the natural pH and microbial diversity of vulvar skin. To our knowledge, this was the first study to assess the effect of an antimicrobial feminine gel wash on the natural pH and vulvar microbiome habitat of the skin using bacterial 16S rRNA and fungal ITS genetic sequencing techniques.

## 1. Introduction

Intimate hygiene products are widely used by women as part of their daily cleansing routine; however, there is a lack of scientific literature about the impact of intimate personal hygiene product use on the vulvar area and even less information is available on the impact on vulvar microbiome stability. Proper feminine hygiene is critical for women’s intimate health, as the vulva is a woman’s first line of defense to protect the genital tract from infection [1]. Vulvar skin is transitional in nature, changing from keratinized skin to non-keratinized mucosal epithelium, and can be differentiated from other skin sites based on many attributes, such as the level of hydration, friction, and permeability [2]. Being covered by a thin stratum corneum with large hair follicles renders vulvar skin to be more susceptible to topical agents and irritation, leading to dermatitis and other dermatologic conditions when its barrier function is affected by factors such as moisture (urine, vaginal discharge), enzymes (stool residue), friction, and heat [2,3]. Symptoms of vulvovaginal disorders are common and can have a significant impact on quality of life.

The microbial inhabitants of the skin are highly dependent on the microenvironment of the biogeographic site and, based on a large-scale body-mapping study, can be grouped into dry, moist, and sebaceous sites [4]. However, not much is known about the transitional vulvar skin, which could be considered a combination of moist and sebaceous sites. In particular, the vast majority of research to date has focused on determining the composition of bacterial communities present in the vaginal orifice of healthy and some symptomatic women. Similar to skin health, the natural microflora of the vulva plays a major role in immunity and maintaining vulvovaginal health by creating a low pH environment that is inhospitable to transient or invading organisms from taking foothold. Although what constitutes a “healthy vulvovaginal microbiome” has not been clearly defined, the concept of dysbiosis is understood as a condition attributed to a microbial imbalance or impaired microbiota where the naturally occurring dominant species are outcompeted by overgrowth of species associated with illness and disorders such as bacterial vaginosis and yeast infection.

A healthy vaginal microbiome is dominated by *Lactobacillus: Lactobacillus iners, Lactobacillus crispatus, Lactobacillus jensenii, and Lactobacillus gasseri* [5]. These organisms support a healthy vaginal environment by mechanisms such as the natural production of lactic acid and maintaining a low pH, secreting bacteriocins, competing for nutrients and receptors, and contributing to innate immunity via hormonal cycling induced by glycogen release and continual sloughing of epithelial cells [6,7,8]. Healthy women can also have non-*Lactobacillus* dominated vaginal and/or vulvar flora, such as *Atopobium, Corynebacterium, Anaerococcus, Peptoniphilus, Prevotella, Mobiluncus, Gardnerella,* and *Sneathia* [6,9,10].

Supported by international guidelines [11,12], daily gentle cleansing of the vulva is important for feminine hygiene and overall intimate health. Routine washing of the vulva prevents the accumulation of vaginal discharge, urine, and fecal contamination to avoid offensive body odor and help maintain healthy vulvar skin for defense against infection. Gentle, proper cleansing care of the vulvar skin is essential, as harsh soaps may irritate this sensitive skin area. Studies of repeated washing of the skin with common alkaline soaps and synthetic detergents have shown that even minor differences in pH of the cleansing products affects both the skin surface pH and bacterial microflora [13,14,15]. While there has been a recent increase in the availability of intimate hygiene products for cleanliness and odor control, some of these products may upset the natural pH of the vulvovaginal area, which may subsequently impact the composition of the normal vulvovaginal microbiota that is needed for protection against infection [16]. In vitro data also suggest that some over-the-counter vaginal products may negatively affect beneficial *Lactobacillus* species in the normal vaginal flora and alter the vaginal immune barrier [16,17,18], which are needed for protection against infection.

Thus, ideal female intimate hygiene products that support intimate health should be specifically formulated and clinically tested for the vulvovaginal area and need to be mild and hypoallergenic, provide protection against dryness, and also maintain the natural pH and microflora. Due to the risks associated with internal washing or douching, external feminine washes are considered more appropriate and may serve as beneficial adjunct therapy in women with vaginal infections or taking antibiotics [19].

A novel gel wash formulation containing lactic acid (pH 4.2) intended for daily external use was formulated to provide gentle cleansing, freshness, and antimicrobial protection to help maintain a healthy balance of the intimate skin area for women. The gel wash contains 2% lactic acid, a key natural antimicrobial ingredient that correlates with vulvovaginal health, as lactic acid helps maintain an acidic pH in the vulvovaginal area to augment skin homeostasis and prevent the growth of harmful bacteria [20,21,22]. The formulation was tested in vitro following standardized ASTM (American Society for Testing and Materials) test methodology (ASTM E2315-03 Standard Guide for Assessment of Antimicrobial Activity Using a Time-Kill Procedure) and demonstrated effective antimicrobial activity (minimum of 1 log kill) against fungi (*Candida albicans*), Gram-positive (including activity against Group B *Streptococcus* and *Staphylococcus aureus*) and Gram-negative bacteria (including *Escherichia coli*; data not shown).

The current clinical study sought to assess, under gynecologist direction, tolerance of this new gel wash when used on the external genital area at least once daily over a 4 week period; the study also evaluated subjective tolerance, skin moisturization, and vulvar skin pH. In addition, this study assessed the impact of the gel wash on the vulvar microbiome by measuring species richness and diversity of the bacterial and fungal communities using genetic sequencing techniques. Respect for the natural pH and microflora of the skin is important for the health of the skin and vulvovaginal area. To our knowledge, this is the first time that the effect of an intimate cleansing product on vulvar skin pH and its associated microbiota has been studied this extensively in a broad age population by performing bacterial 16S rRNA gene and fungal internal transcribed spacer (ITS) microbial diversity analysis on the resident bacterial and fungal communities.

## 2. Results

### 2.1. Subjects

A total of 36 subjects were enrolled in the study—of whom, 34 (all Caucasian) completed the study. Two subjects withdrew after baseline measurement but prior to the Day 14 assessment; therefore, their data were only included in the analysis for skin short-term moisturization baseline (T-BL) to T-1 hour, but were excluded from skin tolerance, skin pH and skin microbiome analysis. Demographic data are summarized in Table 1. Subjects were evenly distributed across the three age groups.

### 2.2. Tolerance/Dermal Irritancy Scores

Total mean irritancy scores increased from 0.01 at Day 0 to 0.12 at Day 14 and 0.13 at Day 28 (Table 2). The mean within-subject change from baseline total irritancy scores at Day 14 and Day 28 were 0.10 and 0.12, respectively, between none and very slight and considered well tolerated. For subjective evaluation of tolerance, 31 (91.18%) subjects at Day 14 and 30 (88.24%) subjects at Day 28 indicated that they did not notice any signs or symptoms and indicated that the test product had been well tolerated by their skin. The rating provided on the global assessment of tolerance was “very good” for 30 (88.2%) subjects, “good” for two (5.9%) subjects, “acceptable” for one (2.9%) subject, and “poor” for one (2.9%) subject; the subject who gave a poor rating reported slight drying and redness and reported using the product more frequently than usual hygiene routine. No subject provided a rating of “very poor”. Based on an overall gynecologist assessment after 28 days of use, this product was considered to be well tolerated by most subjects.

### 2.3. Dryness Scores

The mean within-subject change from baseline dryness scores were 0.03 (upper one-sided confidence limit, 0.08) at Day 14 and 0.01 (upper one-sided confidence limit, 0.04) at Day 28. Of 34 subjects, 33 did not show an increase in dryness from baseline to Day 28, with a lower confidence bound (L95%CL) of 86.8%.

### 2.4. Skin pH Measurements

The mean pH of the overall population was 5.88 at baseline, 5.87 at Day 14, and 5.85 at Day 28, for a mean within-subject change from baseline of −0.01 and −0.03, respectively (Table 3). Statistical analyses confirmed no significant differences in skin pH of the external vulvar area over time with product use (ANOVA; *p* = 0.965).

### 2.5. Skin Moisturization

For short-term (single application) skin moisturization, the mean baseline-control adjusted corneometry ratio for the test product was 1.14 at T-0 immediate and 1.21 at T-1 hour. Analyses showed a significant increase in skin moisturization (baseline-control adjusted corneometry ratio >1.0) at each of the T-0 immediate post-wash and T-1 hour assessments for the test product (*p* < 0.0001; Figure 1A). For the evaluation of longer-term skin moisturization, the mean skin capacitance level was significantly higher at Day 14 (17.44%; *p* = 0.0001) and Day 28 (24.82%; *p* < 0.0001; Figure 1B) compared with baseline.

### 2.6. Adverse Events

There were five adverse events reported during the study—none of which were determined to be product related. These included light headedness, anemia, sinusitis, nausea, and common cold. One subject reported slight drying of the vulvar area (0.5 in a 0–3 dryness scale), which was determined by the gynecologist to have moderate association to the product. The subject also reported using the test product more often than normal hygiene routine.

### 2.7. Microbiome Sequencing and Analysis

Data from 34 subjects were included in the bacterial analysis. Extraction controls and negative PCR controls were all reported as negative after amplification and sequencing (data not shown). Comparing bacterial alpha diversity metrics across time, observed OTUs (Operational Taxonomic Unit) was 222.74, Chao1 Richness (abundance-based estimator of species richness) was 246.64 ± 9.64, and Shannon Diversity (estimator of species richness and species evenness) was 2.58 at baseline; respective values were 176.47, 201.55 ± 10.83, and 2.37 for Day 14, and 186.56, 210.2 ± 10.16, and 2.36 for Day 28. ANOVA test for effect of time on either bacterial species richness or diversity showed no significant change (Table 4). Chao1 Richness and Shannon Diversity within the total bacterial microbiome data colored by time are illustrated in Figure 2. Individual subject bacteria data are attached in Appendix A. Figure 3 shows the relative abundance of the 19 most dominant bacterial genera and species in all samples by time. The most common bacterial communities found to be present at baseline and Days 14 and 28 were *Corynebacterium* spp., *Staphylococcus* spp., *Propionibacterium* spp., and *Lactobacillus* spp. No significant differences were found in any of the top species across three different time points, as shown in Table 5.

Data from 16 subjects were included in the fungal analysis. Subjects were only included if sample amplification was achieved for baseline and at least one more data point. In terms of fungal alpha diversity metrics across time, observed OTUs was 5.06, Chao1 Richness was 5.08 ± 0.08, and Shannon Diversity was 0.37 at baseline; respective values were 4.75, 4.75 ± 0.04, and 0.37 for Day 14, and 6.31, 6.31 ± 0.05, and 0.48 for Day 28. 

ANOVA test for effect of time on either fungal species richness or diversity showed no significant change (Table 6). Figure 4 illustrates Chao1 Richness and Shannon Diversity within the total fungal microbiome data colored by time. Individual subject fungi data are attached in Appendix A. The relative abundance of the 19 most dominant fungal genera and species in all samples by time are shown in Figure 5A,B, respectively; The most common fungal communities found to be present at baseline, Days 14 and 28 were *Malassezia globose*, *Cryptococcus* spp., and *Rhodotorula mucilaginosa*. No significant differences were found in top species across three different time points, as shown in Table 7. 

Differences in beta diversity metrics (unweighted and weighted UniFrac) among age groups overall including all time points were observed (PERMANOVA; *p* < 0.05) for all tests other than weighted UniFrac for fungal data (PERMANOVA; *p* = 0.434); however, no statistical significance was seen among ages in just the baseline sampling, except for within the fungal unweighted UniFrac (PERMANOVA; bacteria unweighted UniFrac *p* = 0.164, bacteria weighted UniFrac *p* = 0.637, fungi unweighted UniFrac *p* < 0.01, fungi weighted UniFrac *p* = 0.554). Relative abundances of bacterial and fungal species for subjects among age groups at baseline depict a vast amount of interindividual variation within age groups (Figure 6A–D).

Correlation analysis was performed between skin pH, and skin moisturization and top taxa to determine whether there are any positive or negative correlations (Figure 7). 

Significant correlations were only found for bacterial but not fungal species after multiple hypothesis correction (FDR). Several *Corynebacterium* species are positively correlated with skin pH, while *Propionibacterium*, *Staphylococcus*, and *Lactobacillus* species are negatively correlated with skin pH. *Firmicutes*, *Clostridiales*, *Clostridia*, *Clostridium*, and *Prevotella timonensis* are negatively correlated with skin moisturization. 

## 3. Discussion

Feminine hygiene is an important component of women’s overall intimate health. Because harsh soaps and surfactants may irritate the sensitive vulvar skin, intimate feminine wash products should be formulated and tested specifically for the vulvar area to ensure they are gentle enough for daily use and also respect the pH and natural microflora of the external genital area. Improper hygiene product usage and hygiene habits can lead to common vulvovaginal disorders [1].

Under gynecologist control, this clinical study evaluated objective and subjective tolerance, pH, skin moisturization, and microbiological diversity of the vulvar area when a new lactic acid-containing gel wash was used at least once daily by women between the ages of 19 and 55 years to wash the external genital area. This new lactic acid-containing gel wash was specifically formulated to achieve a product pH of 4.2 to be compatible with the normal skin pH range and to help maintain vulvovaginal skin homeostasis and provide protection against harmful bacteria. As mentioned above, a healthy vagina is dominated by *Lactobacillus*, a non-sporing, Gram-positive bacillus that produces lactic acid, resulting in a characteristic acidic environment (pH 3–4). The vaginal mucosa is also a rich source of lactic acid, which is produced during estrogen-regulated anaerobic glucose metabolism [23]. Vaginal lactic acid is important, as it correlates with vaginal health by inhibiting the growth of bacteria known to cause vulvovaginal infections [20] and may also play a part in local immune defense [24]. External feminine washes, particularly those containing lactic acid and formulated to an acidic pH that enhances skin homeostasis, are considered more appropriate than internal washes or douches and may be a useful adjunct therapy for women with vaginal infections or taking antibiotics [21]. 

In this clinical study, gynecologist assessment of the vulvar skin during test product use showed acceptable tolerance, as the study population showed no signs of increased dryness, redness, edema, itching, stinging, or burning, and the gynecologist concluded that the product was well tolerated by most women. Previous studies evaluated the effects of sodium lauryl sulfate (SLS, a common surfactant in body wash and feminine wash products) on vulvar skin. While irritant reactions were inconsistent (none in 1 study with low-concentration SLS [0.1–1.0%], but 50% developed irritant dermatitis in another study [SLS 2–5%]), decreases in vulvar skin stratum corneum hydration was observed in both studies [25,26]. Use of the gel wash in this study, however, was associated with significant increases in both short-term (single application) and longer-term (daily use for 28 days) skin moisturization, potentially reducing the possibility of skin dryness and irritation. 

Maintaining the natural pH of the skin is important, as it is integral to the maintenance of the natural microflora, which is essential to protect against invading pathogenic organisms and infections [27,28,29,30]. In the current study with the new gel wash optimally formulated at pH 4.2 to be compatible with healthy skin pH, there was no significant change in vulvar skin pH for the overall population over time with daily product use. Vulvar skin pH in the overall study population ranged from 5.5 to 5.8 from baseline through product use. This is higher than the acidic range (~3.8–4.4) reported for the vagina during reproductive ages, but consistent with the literature, which reports natural skin pH in the acidic range (pH, 4.5–6.0) [27,28,29], and with other studies specifically evaluating vulvar skin pH. In a study of the vulvar skin of healthy Thai women, skin pH was acidic at all sites, but was higher around the vulvar area (labia, mons pubis; pH > 5) than at the other two control sites (inner thigh, forearm; pH ≤ 5) [31]. In a study of healthy Japanese women, skin pH was similar between sites (labia, groin, mons pubis, and inner thigh), with a mean pH of 6.0 [32]. Previous studies showed even small changes in the pH from skin cleansing products may affect skin surface pH and microflora when tested on the forehead and forearm [13]. The current study shows this new lactic acid-containing gel wash optimally formulated to skin pH maintains the vulva skin pH over time after repeated daily washes.

Further, this study demonstrates that this new lactic acid-containing gel wash respects the natural flora of the vulvar skin. Genetic sequencing techniques revealed that the new gel wash did not affect natural microbiome species richness or diversity. Pre- and post-test product use, the predominant bacterial genera were *Corynebacterium*, *Staphylococcus*, *Lactobacillus*, *Actinobacteria*, *Prevotella*, *Clostridia*, and *Propionibacterium* and the predominant fungi were *Malassezia*, *Cryptococcus*, *Rhodoturula*, *Cladosporium*, *Saccharomyces*, and *Penicillium*. As previously discussed, the skin microbiome varies by skin site and physiology, with specific species associated with moist, dry, and sebaceous microenvironments [4], and the vulvar skin, while not extensively studied, likely represents a combination of organisms known to inhabit moist (*Staphylococcus* and *Corynebacterium* spp.), sebaceous (*Propionibacterium* and *Malassezia* spp.), and dry (*Actinobacteria* spp.) areas. The predominant organisms found in our clinical study are similar to those found previously with culture-based techniques using the same detergent scrub method for sampling the vulvar skin (lipophilic and non-lipophilic diphtheroids [including *Corynebacterium*], coagulase negative *Staphylococcus*, micrococci, *Lactobacilli*, *Streptococci*, gram-negative rods, and yeasts) [33]. The percent incidence reported for yeast was 39% for the vulva, consistent with our finding, where 16 out of 34 subjects have shown fungal amplification. Results using non-culture based methods showed a distinctive microbiota for each individual woman tested. Contributing factors that could affect the composition of vulvar communities may include different habits and practices (e.g., frequency of bathing, kinds of clothing worn); microbiota of the labia majora included species found on the skin, including *Staphylococcus* and *Corynebacterium* [9]. In a study of healthy Japanese women, the most predominant species found on the pubic area skin (labia and groin, mons pubis, and inner thigh) were *Lactobacillus* spp. and *Staphylococcus epidermidis*, followed by *S. aureus* [32]. *Propionibacterium acnes* was detected in almost all women at all sites, but was less abundant than *S. aureus*, while *Prevotella* spp. was detected in the labia and groin but not in other sites based on reverse transcription PCR [32].

Particularly noteworthy, despite the antimicrobial action demonstrated in vitro, the gel wash had no significant impact on commensal species richness or diversity of the vulvar skin microbiome; in other words, the wash helped to maintain the natural flora of the external genital area. Grice and Segre [4] discuss that maintaining the delicate balance between the skin, in this case the vulvar skin, and the diverse collection of microorganisms that regularly inhabit the area, is important and can be impacted by endogenous (e.g., genetic variations) and exogenous (e.g., environmental) factors. More specifically, soaps and other hygiene products with high surfactant and high pH have the potential to negatively affect the composition of the skin microbiome. 

Ecological body site niche is a greater determinant of microbiota composition than the individual genetic variation among healthy volunteers. In the current study, a similar microbiota was observed among healthy volunteers for this same site. In a systematic, multi-site metagenomic study of the skin from 15 healthy adults from 18 defined anatomical skin sites, Oh and colleagues [34] concluded that both biogeography and individuality influence skin microbial composition and function.

Interestingly, differences in beta diversity metrics among age groups overall were observed, with substantial interindividual variation in relative abundances of bacterial and fungal species within age groups. At baseline, considerable differences were observed in relative abundance plots at both genera and species level, especially in the fungal community. This could suggest that age is an important factor to influence microbiome composition. However, future studies with more subject numbers in each age group is warranted to confirm this interesting observation.

Correlation analysis was performed between skin pH and moisture levels with the top bacterial taxa, as scientific investigations have shown that applying specific probiotic strains to skin, such as *Staphylococcus epidermidis*, can significantly improve skin moisture retention by increasing the lipid content and suppressing the water evaporation from the skin [35]. Our data suggest that several coryneforms are less abundant in lower pH conditions whereas *Propionibacterium*, *Staphylococcus*, and *Lactobacillus* species thrive better in a lower pH environment. Predominantly the *Clostridia* and *Clostridium* species showed a decrease with increase in moisturization. 

A healthy vulvovaginal environment is particularly important for a pregnant woman to protect the baby from potential infection risk. Protection against Group B *Streptococcus* is particularly important for pregnant women as it often colonizes the vagina through the gastrointestinal tract and increases the risk of preterm delivery, neonatal meningitis, and even fetal death. The association of harmful vaginal bacteria, i.e., bacterial vaginosis, with preterm labor and preterm birth is also supported by the literature [1]. Birth mode (cesarean vs. vaginal birth) can have long-term effects on microbial diversity [36], but data suggest that the skin microbiome of a cesarean-born infant can be at least partially restored through vaginal microbial transfer to what would be expected following vaginal birth [37]. In contrast, Chu and colleagues showed that while the infant microbiome matures and changes over the first 6 weeks of life, the variations observed are primarily due to body site and not mode of delivery [38]. Despite inconclusive data, pregnant women should be cautious in choosing hygiene products because they might impact the early-life microbiome, which is critical for development of infant immune function [39].

Results from the current study should be considered in relation to the study limitations. First, there were no previous vulvar microbiome data to power the study, although data from another internal study assessing the effects of a cosmetic product on the skin pH were used to determine an appropriate subject sample size. The sample size selected for this study meets the requirements of CLT in order to sufficiently sample the population. Second, bacterial 16S rRNA gene and fungal ITS sequencing is limited to reference sequences that are present in the database and low sequence homology within those regions to allow for differentiation between taxonomic groups versus shotgun metagenomics or metatranscriptomics that can also elucidate function. Finally, this was a purely Caucasian population and was geographically limited to the United Kingdom. An additional study using shotgun metagenomics and/or metatranscriptomics including different subjects of different ethnicities is warranted to obtain complete information. Further study specific to postmenopausal women may also be of interest, as vulvovaginal dryness and infections can become increasing concerns for this subgroup of women.

## 4. Materials and Methods 

### 4.1. Eligibility Criteria

Eligible subjects included healthy females in 3 equally balanced age groups spanning from teenage years through menopause (approximately 10–12 subjects per group): aged 18 to 29 years, aged 30 to 44 years, and aged 45 to 55 years (and having ≥2–3 menopausal symptoms and signs as determined by the investigator). Participants were not permitted to apply any skin products other than the gel wash on the external genital area or change from their usual brand of sanitary products and/or laundry detergent for the duration of the study. Subjects were excluded if they were pregnant; post-menopause; had active psoriasis, eczema, or other active skin disorder; had dryness in the external genital area; were presently or had taken antibiotics within the past 3 months; or had changed contraceptive medication or menopausal/hormonal treatment in the past 3 months or intended to change treatment during the study period.

All experimental procedures were performed in compliance with the Reading Independent Ethics Committee (RIEC Ref. 130115-3, approved on 2 February 2015). In addition, written informed consent was obtained from all volunteers in this study and was submitted to the related ethics committee.

### 4.2. Study Design and Procedures

This was an open, uncontrolled study conducted under gynecologist control. During the 7 day pre-study conditioning period, subjects were instructed to replace their usual body/shower wash with a commercially available shower gel product (pH 5.6) to minimize the impact from personal care products such as harsh high pH soaps and wash their entire body with the product at least once per day, including the external genital/intimate area. After the 7 day pre-wash period, subjects used the gel wash (i.e., test product; pH 4.2) to wash the external genital area (mons pubis and vulva) and entire body at least once per day for 28 days. The test product could be used as often as needed but was not to be applied internally into the vagina. The test product had to be applied with the hands only (no washcloths, poofs, etc.), and each use was recorded in a subject diary to determine compliance. Subjects were instructed not to shower/bathe the morning of study visits (Days 0, 14, and 28) and not to wash their external genital area (including the vulvar area) with the pre-wash product or the test product for at least 8 hours but not more than 16 hours before their assessment visit at the study center.

### 4.3. Study Assessments and Endpoints

#### 4.3.1. Skin Tolerance

Skin tolerance of the test product was assessed by a gynecologist at baseline and at Days 14 and 28. Tolerance assessments included skin dryness, erythema, edema, desquamation, itching (verbal questioning of subjects, visual assessment for scratch marks), burning (verbal questioning of subjects), stinging (verbal questioning of subjects), and other clinical signs of irritancy of the genital area; results were recorded using a 5-point scale: 0 = “none”, 0.5 = “very slight”, 1 = “slight”, 2 = “moderate”, and 3 = “severe”. A global assessment of tolerance was also made by the gynecologist for each subject at Day 28 using a 5-point scale: 1 = “very good”, 2 = “good”, 3 = “acceptable”, 4 = “poor”, and 5 = “very poor”. At the end of the study, the gynecologist made an overall summary assessment of test product tolerance as follows: *The test product was very well/well/moderately well/not well/not at all well tolerated by all/most/some/a few subjects*. Subjects also provided subjective tolerance assessments at baseline and at Days 14 and 28. Adverse events were also recorded throughout the study.

#### 4.3.2. Skin pH

Skin pH of the external vulvar area (mid-labium majus) was measured at baseline and at Days 14 and 28 with a Multi Probe Adaptor System using the skin-pH meter PH 905 probe (Figure 8). Skin pH measurements were taken prior to microbiological sample collection and with the pH probe within the circumference of the microbiological sampling site. Two measurements were taken: 1 at the anatomical left side and 1 at the anatomical right side of the external vulvar skin area; the arithmetic mean of the 2 replicate pH measurements was calculated. For each assessment (Days 0, 14, and 28), the arithmetic mean of the total population was calculated. To explore potential pH changes in the overall population over time, an analysis of variance (ANOVA) model that included subject and time was carried out at each time point. Least squares means with confidence intervals were calculated at each time point. Short-term (single application) and longer-term (daily use) skin moisturization measurements of the upper inner leg and groin areas, respectively, were utilized to avoid interference with pH and microbiome assessments (Figure 8). The short-term effect of a single application on skin moisturization was measured in the upper inner leg area at 2 locations (control and test product application site) before and at 2 time points after application of the test product by the study technician using a Corneometer (Corneometer^®^ CM 825 (Courage & Khazaka GmbH) at baseline (T-BL), T-0 post wash (immediately after test product wash), and T-1 hour after application of test product wash after the subject had remained in climate-controlled conditions for 20 minutes following baseline assessments. Three replicate measurements were taken from both test sites for each assessment time and averaged (arithmetic mean) to give a single value for each subject, site, and assessment. The longer-term effect of daily use of the test product on skin moisturization was measured in the groin area using a Corneometer^®^ SM 825 at baseline and at Days 14 and 28 after the subject had remained in climate-controlled conditions for 20 minutes. Three replicate measurements were taken on each occasion and averaged (arithmetic mean) to give a single value for each subject and assessment. 

#### 4.3.3. Sample Collection and Microbiome Analysis

Microbiological samples were taken from the mid-labium majus area (both sides; Figure 8), including both keratinized and mucosal skin, via a modified liquid cup scrub method [40] at baseline (Day 0) and Days 14 and 28. A sterile glass cylinder (approximately 4 cm^2^ in area) was pressed to firmly make contact with the skin exerting adequate pressure to prevent leaking of liquid during collection. One milliliter of 0.1% detergent solution (75 mM phosphate buffer containing 0.1% [v/v] Triton X-100 buffered to a pH between 7.5 and 8.0) was pipetted into the sterile glass cylinder circumscribing an area of about 3.0 cm^2^, and the skin surface was carefully rubbed as evenly as possible with a blunted, sterile, glass rod (scrubbing device), applying moderate constant pressure for 1 minute. Samples taken from both sides were pooled for the microbial diversity analysis. Samples were heat inactivated and shipped to RTL Genomics, a division of Research and Testing Laboratory (Lubbock, TX) for microbial diversity sequencing.

Whole genomic DNA was extracted from each sample using a MoBio PowerMag Soil DNA Isolation Kit (optimized for Kingfisher; MoBio Laboratories, Carlsbad, CA). DNA was amplified for both bacterial 16S rRNA gene and fungal ITS sequencing in a 2-step process. The forward primer was constructed with (5′-3′) the Illumina i5 sequencing primer (TCGTCGGCAGCGTCAGATGTGTATAAGAGACAG) and the 28F primer (GAGTTTGATCNTGGCTCAG) for bacterial analysis and the ITS1F primer (CTTGGTCATTTAGAGGAAGTAA) for fungal analysis. The reverse primer was constructed with (5′-3′) the Illumina i7 sequencing primer (GTCTCGTGGGCTCGGAGATGTGTATAAGAGACAG) and the 388R primer (TGCTGCCTCCCGTAGGAGT) for bacterial analysis and the ITS2aR primer (GCTGCGTTCTTCATCGATGC) for fungal analysis. Amplifications were performed in 25 µL reactions with Qiagen HotStar Taq master mix (Qiagen Inc, Valencia, CA), 1 µL of 5 uM primer mix, and 1 µL of template. Reactions were performed on ABI Veriti thermocyclers (Applied Biosystems, Carlsbad, CA) under the following thermal profile: 95°C for 5 minutes, then 25 cycles of 94°C for 30 seconds, 54°C for 40 seconds, 72°C for 1 minute, followed by 1 cycle of 72°C for 10 minutes and 4°C hold. Products from the first stage amplification were added to a second polymerase chain reaction (PCR) based on qualitatively determined concentrations. Primers for the second PCR were designed based on the Illumina Nextera PCR primers as follows: Forward—AATGATACGGCGACCACCGAGATCTACAC[i5index]TCGTCGGCAGCGTC and Reverse—CAAGCAGAAGACGGCATACGAGAT[i7index]GTCTCGTGGGCTCGG. The second stage amplification was run the same as the first stage except for 10 cycles. Amplification products were visualized with eGels (Life Technologies, Grand Island, NY). Products were then pooled equimolar and each pool was size selected in 2 rounds using Agencourt AMPure XP (BeckmanCoulter, Indianapolis, IN) in a 0.7 ratio for both rounds. Size selected pools were then quantified using the Quibit 2.0 fluorometer (Life Technologies) and loaded on an Illumina MiSeq (Illumina, Inc., San Diego, CA) 2x300 flow cell at 10 pM. Extraction controls and negative PCR controls were included throughout the extraction, amplification, and subsequent sequencing process. 

The data analysis pipeline consisted of 2 major stages, the denoising and chimera detection stage and the microbial diversity analysis stage. During the denoising and chimera detection stage, denoising was performed using various techniques to remove short sequences, singleton sequences, and noisy reads. With the bad reads removed, chimera detection was performed to aid in the removal of chimeric sequences. Lastly, remaining sequences were then corrected base by base to help remove noise from within each sequence. During the diversity analysis stage, each sample was run through RTL Genomic’s analysis pipeline to determine the taxonomic information for each constituent read using RTL Genomic’s in-house curated reference database, and then this information was collected for each sample yielding operational taxonomic units (OTUs) and taxonomic level information. 

To assess if there were within-sample changes that differed significantly over time, overall species richness (Chao1 Richness) and species diversity (Shannon Diversity) were calculated using the phyloseq package in R [41]. From these alpha diversity metrics, ANOVA tests were performed in R to assess if there were any significant differences across time points. To examine microbial community compositions, genus- and species-level OTU relative abundance, bar plots were generated using ggplot2 in R [42], and ANOVA tests with multiple hypothesis correction (FDR) were performed to assess if there were any significant differences across time points. 

To assess correlation of pH and moisturization measurements with species relative abundance, Spearman correlations with multiple hypothesis correction (FDR) were performed.

Since no similar studies have been performed previously, 36 subjects were sampled in this study in order to meet the requirements of central limit theorem (CLT) to sufficiently sample the population.

## 5. Conclusions

The objective was to develop a gel wash formulation that not only provides mild cleansing care but also maintains the health of the intimate skin barrier. This was the first study to assess the effect of a feminine gel wash on the natural pH and the vulvar microbiome habitat of the skin using bacterial 16S rRNA and fungal ITS genetic sequencing techniques. The study showed that this new gel wash is a mild, moisturizing cleanser that does not harm and instead respects the natural pH and microbial diversity of the intimate skin. This study examined the impact of a novel physiologically relevant pH feminine wash product on the vulvar microbial communities prior to and following product use over time for up to 1 month in three distinct age groups. Based on this analysis, we now have a better understanding of the bacterial and fungal microbial communities that inhabit the external vulvar area in healthy women and have established the foundation for future studies looking to investigate the stability and impact of product use in the vulvovaginal area on the vulvar microbiome. The gel wash formulation we assessed demonstrated no significant impact on both the species richness and diversity. As respect for the natural pH and microbiome of the skin is important for the health of the intimate skin and vulvovaginal area, evaluating the microbiome of the skin should be considered standard practice when evaluating feminine products clinically for their effects on the skin.

## Figures and Tables

**Figure 1 antibiotics-09-00055-f001:**
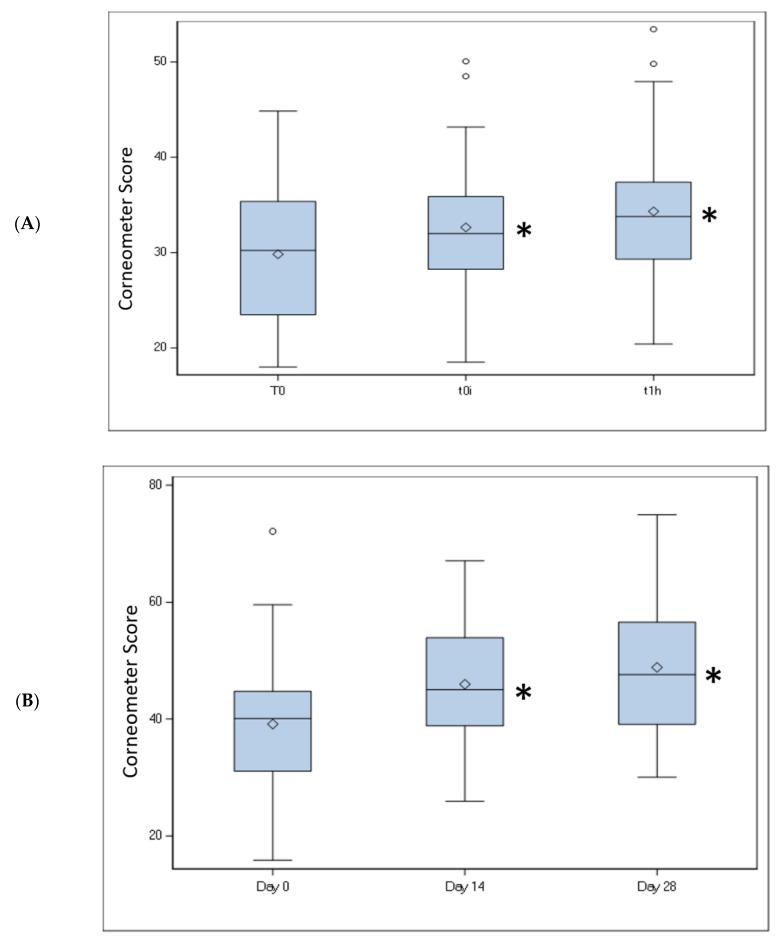
Skin moisturization of the intimate area. (**A**) Short-term (single application) mean upper thigh skin moisturization Corneometer score at T-0 immediate and T-1 hour. **p* < 0.0001 vs. baseline. (**B**) Longer-term (daily use) mean groin skin moisturization Corneometer score at Day 14 and Day 28 compared with baseline. **p* < 0.0001 vs. baseline.

**Figure 2 antibiotics-09-00055-f002:**
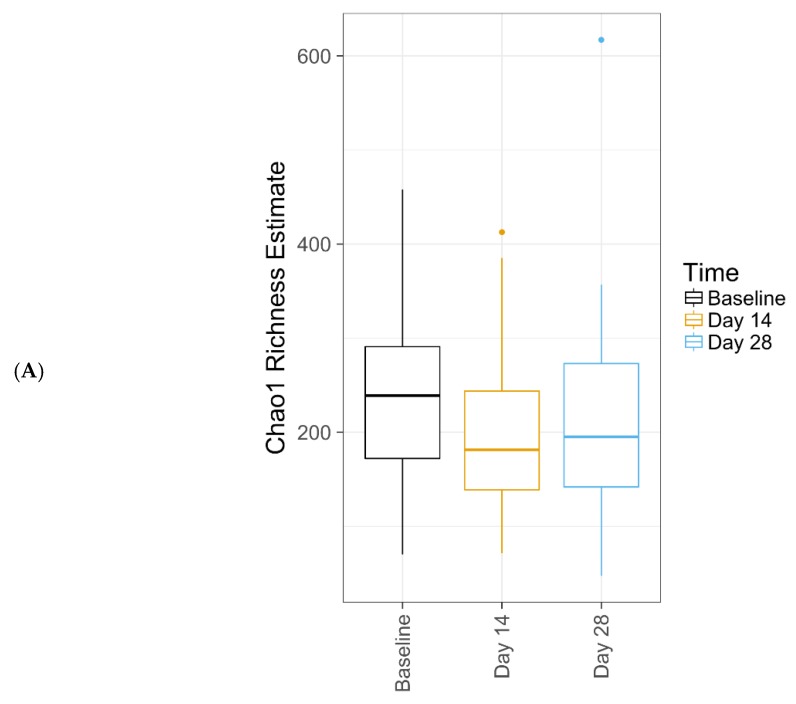
Chao1 Richness (**A**) and Shannon Diversity (**B**) within the total bacterial microbiome data colored by time. The median value (bar), first and third quartiles (hinges), and 1.5 * IQR (whiskers) in each group also are illustrated. Individual data are included in Appendix A.

**Figure 3 antibiotics-09-00055-f003:**
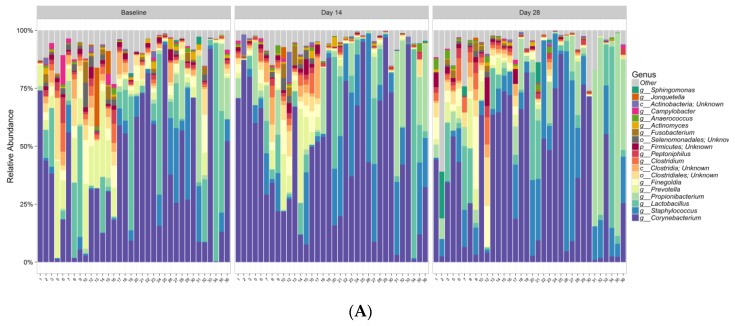
Relative abundance of the 19 most dominant bacterial genera (**A**) and species (**B**) in all samples by time.

**Figure 4 antibiotics-09-00055-f004:**
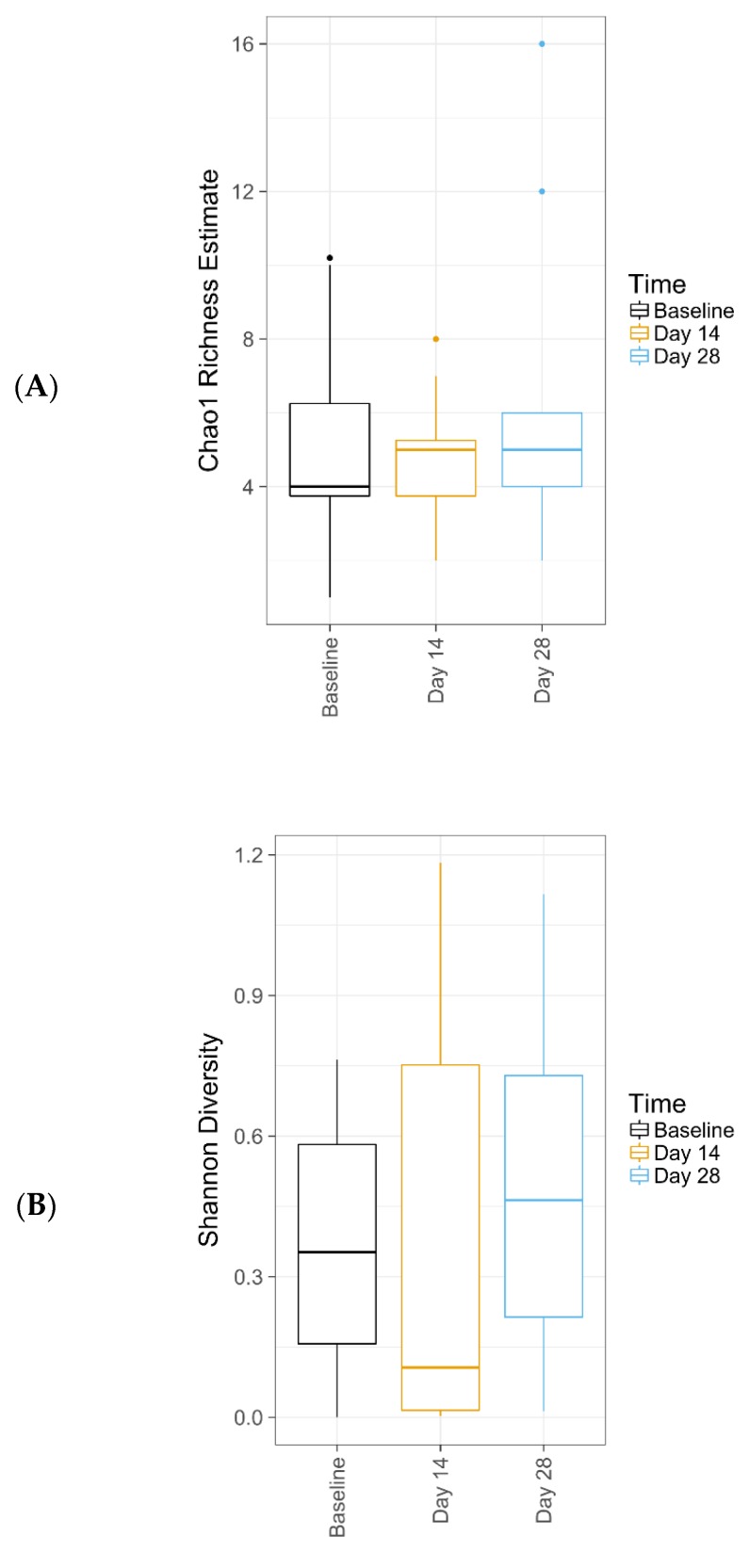
Chao1 Richness (**A**) and Shannon Diversity (**B**) within the total fungal microbiome data colored by time. The median value (bar), first and third quartiles (hinges), and 1.5 * IQR (whiskers) in each group also are illustrated. Individual data are included in Appendix A.

**Figure 5 antibiotics-09-00055-f005:**
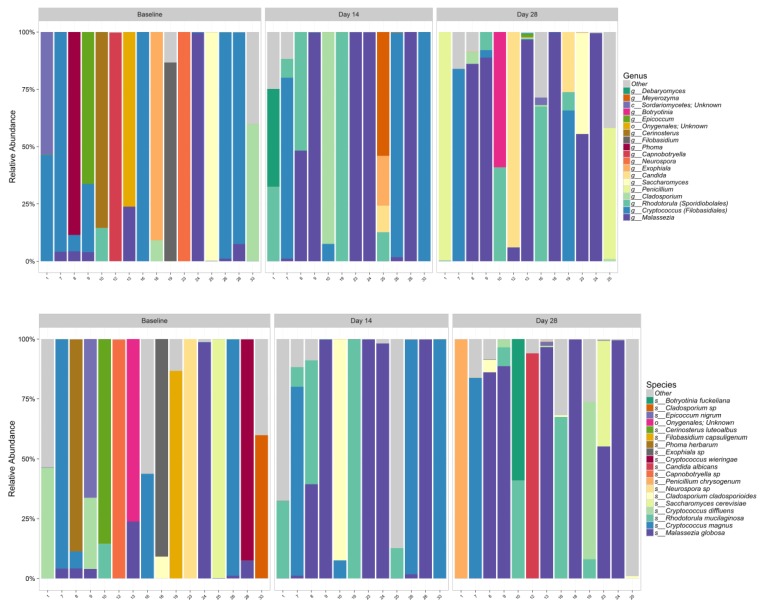
Relative abundance of the 19 most dominant fungal genera (**A**) and species (**B**) in all samples by time.

**Figure 6 antibiotics-09-00055-f006:**
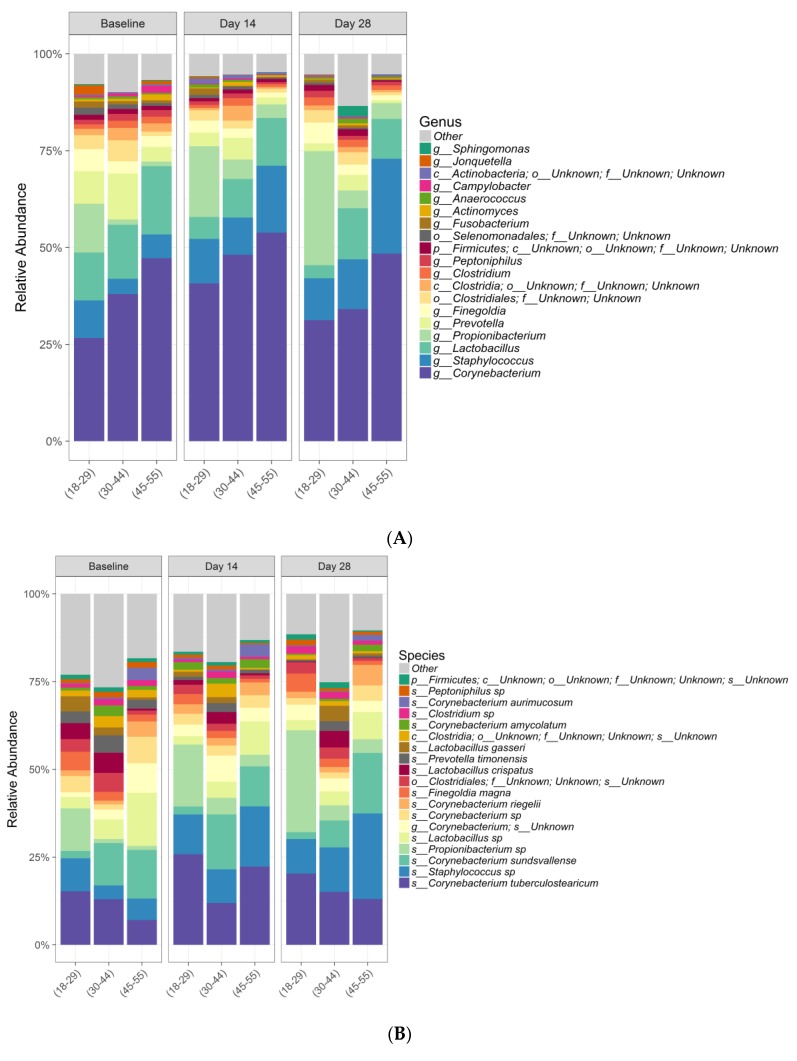
Relative abundance of the 19 most dominant bacteria genera (**A**) and species (**B**) in all samples faceted by age. Relative abundance of the 19 most dominant fungi genera (**C**) and species (**D**) in all samples faceted by age.

**Figure 7 antibiotics-09-00055-f007:**
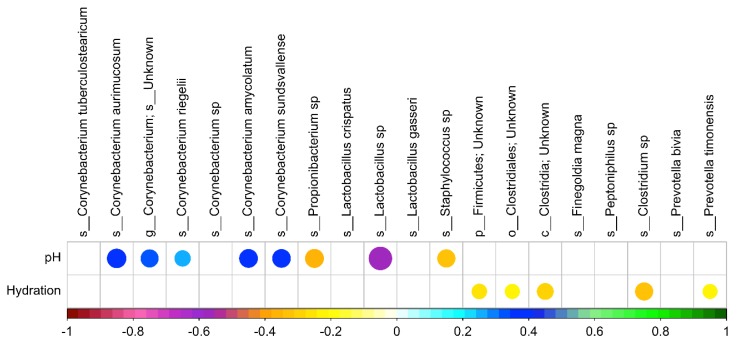
Correlation analysis between skin pH and skin moisturization with top bacterial species. Circles correlate negatively or positively according to color scale. Circle size correlates to level of significance (larger circumference depicts stronger significance).

**Figure 8 antibiotics-09-00055-f008:**
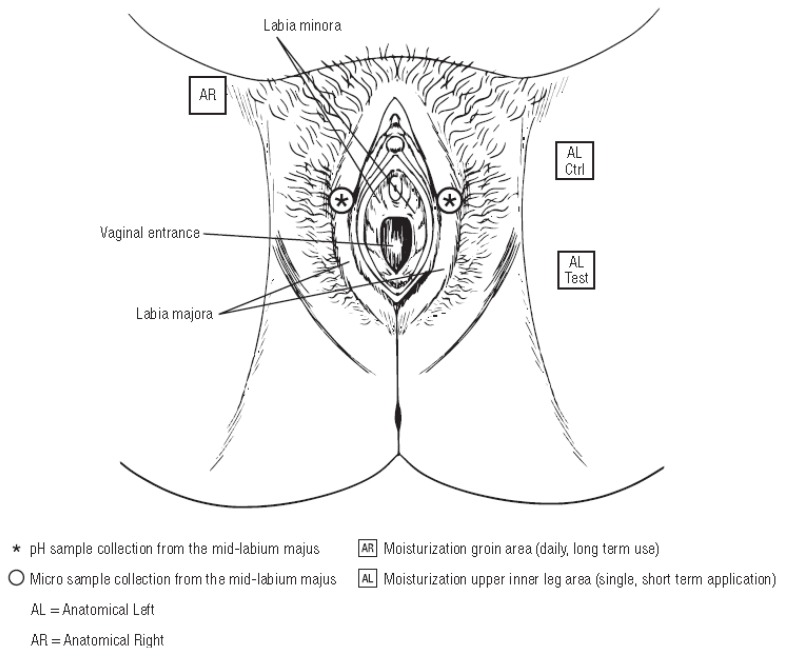
Assessments of the vulva.

**Table 1 antibiotics-09-00055-t001:** Subject demographics.

Characteristic	*n* = 36
Mean (SD) age, y	36.61 (11.28)
Range	19–55
Age group, n (%)	
Age 18–29 years	12 (33.3)
Age 30–44 years	13 (36.1)
Age 45–55 years	11 (30.6)

SD, standard deviation.

**Table 2 antibiotics-09-00055-t002:** Dermal irritancy scores (*n* = 34).

	Assessment		
Mean (SE) score	Day 0	Day 14	Day 28
Dryness	0.00 (0.00)	0.03 (0.03)	0.01 (0.02)
Erythema	0.00 (0.00)	0.06 (0.06)	0.09 (0.07)
Edema	0.00 (0.00)	0.00 (0.00)	0.00 (0.00)
Desquamation	0.00 (0.00)	0.00 (0.00)	0.00 (0.00)
Itching	0.00 (0.00)	0.03 (0.03)	0.03 (0.03)
Scratches	0.00 (0.00)	0.00 (0.00)	0.00 (0.00)
Burning	0.00 (0.00)	0.00 (0.00)	0.00 (0.00)
Stinging	0.01 (0.02)	0.00 (0.00)	0.00 (0.00)
Other	0.00 (0.00)	0.00 (0.00)	0.00 (0.00)
Total irritancy score	0.01 (0.02)	0.12 (0.12)	0.13 (0.09)
Change from baseline		0.10 (0.12)	0.12 (0.09)

SE, standard error.

**Table 3 antibiotics-09-00055-t003:** Skin pH measurement of the vulvar area.

	Skin pH Assessment		
	Day 0	Day 14	Day 28
**N**	34	34	34
**Mean**	5.88	5.87	5.85
**SD**	0.51	0.45	0.47

*p* = 0.965.

**Table 4 antibiotics-09-00055-t004:** Results of the ANOVA testing for differences in Chao1 Richness (abdundance-based estimator of species richness) and Shannon Diversity (estimator of species richness and species evenness) across time: bacterial results.

	Df	Sum Sq	Mean Sq	F Value	Pr (>F)
Chao1 Richness					
Time	2	38,932.63	19,466.32	2.01	0.1400
Residuals	99	960,760.90	9704.66		
Shannon Diversity					
Time	2	1.08	0.54	1.19	0.3097
Residuals	99	44.90	0.45		

**Table 5 antibiotics-09-00055-t005:** Relative abundance of top bacterial species across time.

	Baseline	Day_14	Day_28	Fval	*p*	pjust
*s__Corynebacterium tuberculostearicum*	0.1182	0.1978	0.1614	1.8126	0.1686	0.5927
*s__Staphylococcus sp*	0.0640	0.1259	0.1553	3.3001	0.0410	0.4886
*s__Corynebacterium sundsvallense*	0.0942	0.0994	0.0891	0.0367	0.9640	0.9640
*s__Propionibacterium sp*	0.0469	0.0845	0.1219	1.8111	0.1688	0.5927
*s__Lactobacillus sp*	0.0793	0.0548	0.0484	0.3254	0.7230	0.8684
*g__Corynebacterium; s__Unknown*	0.0412	0.0494	0.0377	0.1803	0.8353	0.9281
*s__Corynebacterium sp*	0.0447	0.0317	0.0261	1.0384	0.3579	0.7157
*s__Corynebacterium riegelii*	0.0231	0.0281	0.0305	0.1209	0.8862	0.9329
*s__Finegoldia magna*	0.0323	0.0200	0.0288	0.6330	0.5331	0.8684
*o__Clostridiales; Unknown*	0.0345	0.0190	0.0242	1.3046	0.2759	0.6897
*s__Lactobacillus crispatus*	0.0366	0.0179	0.0176	0.8489	0.4310	0.7836
*s__Prevotella timonensis*	0.0365	0.0154	0.0132	3.1127	0.0489	0.4886
*s__Lactobacillus gasseri*	0.0242	0.0107	0.0183	0.3478	0.7071	0.8684
*c__Clostridia; Unknown*	0.0235	0.0170	0.0106	1.7414	0.1806	0.5927
*s__Corynebacterium amycolatum*	0.0169	0.0205	0.0095	1.0680	0.3476	0.7157
*s__Clostridium sp*	0.0156	0.0117	0.0171	0.4290	0.6524	0.8684
*s__Corynebacterium aurimucosum*	0.0136	0.0146	0.0073	0.3046	0.7381	0.8684
*s__Peptoniphilus sp*	0.0145	0.0082	0.0112	1.5981	0.2074	0.5927
*p__Firmicutes; Unknown*	0.0121	0.0089	0.0121	0.3619	0.6973	0.8684
*s__Prevotella bivia*	0.0174	0.0066	0.0033	2.5730	0.0814	0.5427

**Table 6 antibiotics-09-00055-t006:** Results of the ANOVA Testing for differences in Chao1 Richness and Shannon Diversity across time: fungal results.

	Df	Sum Sq	Mean Sq	F Value	Pr (>F)
Chao1 Richness					
Time	2	17.44	8.72	0.90	0.4169
Residuals	38	369.97	9.74		
Shannon Diversity					
Time	2	0.11	0.06	0.44	0.6469
Residuals	38	4.92	0.13		

**Table 7 antibiotics-09-00055-t007:** Relative abundance of top fungal species across time.

	Baseline	Day_14	Day_28	Fval	P	pjust
*s__Malassezia globosa*	0.0897	0.3667	0.4046	2.7353	0.0777	0.5279
*s__Cryptococcus magnus*	0.1535	0.2369	0.0645	0.8280	0.4447	0.5279
*s__Rhodotorula mucilaginosa*	0.0091	0.1710	0.0957	2.1956	0.1252	0.5279
*s__Cryptococcus diffluens*	0.0476	0.0000	0.0533	0.6187	0.5440	0.5726
*s__Saccharomyces cerevisiae*	0.0624	0.0000	0.0340	0.4561	0.6372	0.6372
*s__Cladosporium cladosporioides*	0.0057	0.0770	0.0055	1.0380	0.3640	0.5279
*s__Neurospora sp*	0.0625	0.0000	0.0000	0.7721	0.4692	0.5279
*s__Penicillium chrysogenum*	0.0000	0.0000	0.0768	1.0812	0.3494	0.5279
*s__Capnobotryella sp*	0.0623	0.0000	0.0000	0.7731	0.4687	0.5279
*s__Candida albicans*	0.0000	0.0000	0.0723	1.0822	0.3491	0.5279
*s__Cryptococcus wieringae*	0.0577	0.0000	0.0000	0.7721	0.4692	0.5279
*s__Exophiala sp*	0.0568	0.0000	0.0005	0.7661	0.4719	0.5279
*s__Phoma herbarum*	0.0554	0.0000	0.0000	0.7737	0.4684	0.5279
*s__Filobasidium capsuligenum*	0.0542	0.0000	0.0000	0.7717	0.4693	0.5279
*s__Cerinosterus luteoalbus*	0.0534	0.0000	0.0000	0.7728	0.4689	0.5279
*o__Onygenales; Unknown*	0.0476	0.0001	0.0000	0.7711	0.4696	0.5279
*s__Epicoccum nigrum*	0.0415	0.0000	0.0010	0.7590	0.4751	0.5279
*s__Cladosporium sp*	0.0373	0.0000	0.0001	0.7706	0.4699	0.5279
*s__Botryotinia fuckeliana*	0.0000	0.0000	0.0454	1.0813	0.3494	0.5279
*s__Penicillium sp*	0.0000	0.0000	0.0440	1.0831	0.3488	0.5279

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
