# Peer review of "A 28 Day Clinical Assessment of a Lactic Acid-containing Antimicrobial Intimate Gel Wash Formulation on Skin Tolerance and Impact on the Vulvar Microbiome"

_antibiotics, 2020, doi:10.3390/antibiotics9020055_

Round 1

Reviewer 1 Report

This manuscript describes a 4-week clinical study for the assessment of a gel wash containing lactic acid 20 (pH 4.2) when used on the external genital area. The work also involves the study of the effects of the gel on skin moisturization, vulvar skin pH, and the vulvar microbiome. The work is interesting and carefully performed, however, there are a few points which need to be addressed by the authors:

1) The introduction is too long and could be condensed, summarizing the most important aspect of the area.

2) How many replicate measurements of pH were taken from the test sites ?

3) The paragraph ‘Adverse Events’ should be written and discussed in more detail.  

4) The text in the Results Section has been written in italic type fond. Please use normal fond.

5) The correlation analysis between skin pH, skin moisturization with top bacterial species should be discussed and compared with relevant studies of human microbiome.

Reviewer 2 Report

This manuscript is clearly written and the data support the conclusions. I have a couple minor concerns/suggestions as well as some major ones.

Minor Concerns:

Figure 1 is unclear. The X and Y axes in both panels A and B are not properly labeled. Which corneometer instrument was used? (The manufacturer should be mentioned.) Below both Panels A and B it says “*p<0.0001,” however there is no asterisk in the figure in either panel A or B. Finally, how was the statistical analysis done??

In the discussion, the authors make the point, correctly, that altered vaginal bacteria in a pregnant woman can be harmful to the baby. Please also comment on the association of harmful vaginal bacteria, i.e. bacterial vaginosis, with preterm labor and preterm birth, which is supported by a vast amount of literature.

Major Concerns:

The product tested in this study was not compared with the current gold standard product. In other words, although the authors mention that some products currently available may cause irritation, etc., it is not at all clear to me that other existing mild soaps or similar products, which may be less expensive, would also cause no irritation and no change in the vaginal microbiota.

It is true that the product tested in this study generally did not cause any unfavorable effects, such as dryness, irritation of dysbiosis. But the product also did not produce any favorable effects, such as increases in Lactobacillus or decreases in harmful organisms. There appears to have been no advantage to this product as compared to whatever products the participants were using before entering the study, as is underlined by the absence of changes over the course of the trial.

Reviewer 3 Report

This manuscript assessed the effect of using a novel gel wash containing lactic acid (pH 4.2) on the external female genital area including skin tolerance, skin moisturization, vulvar skin pH, and its microbiome stability for both bacteria and fungi.

This manuscript is very well written with a thorough introduction and discussion. The results are well presented in detail.

Noticed the study subjects were required to have a 7-day pre-study conditioning period, with a commercially available shower gel product (pH 5.6). Could you please include the explanation of the rationale and benefit of this pre-study period and the reason to choose this shower gel product in the relevant Method section? Thanks.

During skin microbiome sample collection, one millilitre of 0.1% detergent solution (75 mM phosphate buffer containing 0.1% [v/v] Triton X-100 buffered to a pH between 7.5-8.0) was used in contact with 3 cm2 vulvar skin for 1 minute, is there any evidence from literature that this sampling detergent not affecting skin microbiota? Thanks.

Round 2

Reviewer 1 Report

The manuscript was improved and can be accepted for publication.

Reviewer 2 Report

Although I still feel the absence of a "gold standard" positive control is a weakness of the manuscript, all of my specific comments have now been addressed.